# Antioxidant Peptides from Sacha Inchi Meal: An In Vitro, Ex Vivo, and In Silico Approach

**DOI:** 10.3390/foods13233924

**Published:** 2024-12-05

**Authors:** Erwin Torres-Sánchez, Iván Lorca-Alonso, Sandra González-de la Fuente, Blanca Hernández-Ledesma, Luis-Felipe Gutiérrez

**Affiliations:** 1Facultad de Ciencias Agrarias, Universidad Nacional de Colombia Sede Bogotá, Carrera 30 No. 45-03, Bogotá 111321, Colombia; egtorressa@unal.edu.co; 2Centro de Biología Molecular Severo Ochoa (CBM) Consejo Superior de Investigaciones Científicas (CSIC)-Universidad Autónoma de Madrid (UAM), Nicolás Cabrera 1, 28049 Madrid, Spain; ivan.lorca@cbm.csic.es (I.L.-A.); sandra.g@cbm.csic.es (S.G.-d.l.F.); 3Instituto de Investigación en Ciencias de la Alimentación (CIAL), CSIC-UAM, Campus de Excelencia Internacional (CEI)-(UAM+CSIC), Nicolás Cabrera 9, 28049 Madrid, Spain; 4Instituto de Ciencia y Tecnología de Alimentos (ICTA), Universidad Nacional de Colombia Sede Bogotá, Carrera 30 No. 45-03, Edificio 500A, Bogotá 111321, Colombia

**Keywords:** *Plukenetia volubilis*, INFOGEST, mass spectrometry, bioactive peptides, bioinformatics, antioxidant peptides, bioavailability

## Abstract

Plant-derived antioxidant peptides safeguard food against oxidation, helping to preserve its flavor and nutrients, and hold significant potential for use in functional food development. Sacha Inchi Oil Press-Cake (SIPC), a by-product of oil processing, was used to produce Sacha Inchi Protein Concentrate (SPC) in vitro, hydrolyzed by a standardized static INFOGEST 2.0 protocol. This study aimed to integrate in vitro, ex vivo, and in silico methods to evaluate the release of antioxidant peptides from SPC during gastrointestinal digestion. In vitro and ex vivo methods were used to investigate the antioxidant potential of SPC digests. Bioinformatics tools (find-pep-seq, AnOxPP, AnOxPePred-1.0, PepCalc, MLCPP 2.0, Pasta 2.0, PlifePred, Rapid Peptide Generator, and SwissADME) were employed to characterize antioxidant peptides. The gastric and intestinal digests exhibited higher ABTS and ORAC values than those of SPC. Under basal conditions, gastric digest fractions GD1, GD2, and GD3 (<3, 3–10, and >10 kDa, respectively), separated by ultrafiltration, significantly reduced the ROS levels in the RAW264.7 macrophages while, under LPS stimulation, GD1 (16 µg/mL) and GD2 (500 and 1000 µg/mL) reversed the induced damage. From the de novo peptidome determined, 416 peptides were selected based on their resistance to digestion. Through in silico tools, 315 resistant peptides were identified as antioxidants. Despite low predicted bioavailability, the peptides SVMGPYYNSK, EWGGGGCGGGGGVSSLR, RHWLPR, LQDWYDK, and ALEETNYELEK showed potential for extracellular targets and drug delivery. In silico digestion yielded the sequences SVMGPY, EW, GGGGCGGGGGVSS, PQY, HGGGGGG, GGGG, HW, and SGGGY, which are promising free radical scavengers with increased bioavailability. However, these hypotheses require confirmation through chemical synthesis and further validation studies.

## 1. Introduction

In the body tissues, aerobic metabolism comprises a series of biochemical reactions aimed at converting raw materials into energy and synthesizing macromolecules, releasing reactive oxygen species (ROS) through metabolic pathways. ROS constitute highly reactive and unstable molecules due to unpaired electrons, such as hydrogen peroxide (H_2_O_2_), hydroxyl radicals (HO^●^), peroxynitrite (ONOO), superoxide anion (O_2_^●−^), and peroxyl radicals (ROO^●^) [1]. To counteract ROS production, cells have antioxidant defense mechanisms that convert free radicals into non-reactive species, which are then transformed into harmless components. However, when ROS overload the body’s defenses, oxidative stress occurs, damaging vital cellular components such as proteins, lipids, and DNA [2]. Oxidative stress in the human body increases the risks of developing non-communicable diseases (NCDs), including cancer, inflammation, and cardiovascular and neurodegenerative diseases [3].

Antioxidant peptides can eradicate ROS and impede the reactions mediated by free radicals. Plant-derived antioxidant peptides effectively inhibit lipid peroxidation, preserving food stability and nutrients. They are easily absorbed by the intestinal tract, maintaining their structure to enter the circulatory system and fully exert their antioxidant functions, making them promising candidates for functional food development [4]. Antioxidant peptides offer the additional benefits of being minimally toxic or non-toxic, are abundantly derived from food sources, and are preferred by individuals as an oral preventive strategy over intravenous administration [5]. Food-derived peptides have been extensively researched over the past two decades for their capacity to mitigate oxidative stress-associated NCDs [6,7]. Over 1000 antioxidant peptides have been identified from milk, and animal- and plant-derived foods through enzymatic hydrolysis, fermentation, or autolysis techniques [8]. There are numerous reports regarding peptides derived from by-products. For example, the peptides QFLLAGR, ASPKPSSA, and RGQVIYVL, with angiotensin-I converting enzyme inhibitory and antioxidant potential have been identified in quinoa (*Chenopodium quinoa* Wild.) bran [9]. The peptides IEFL, RYLL, and SAADL were isolated and purified from corn (*Zea mays*) gluten meal, showing good free radical scavenging activity and a lipid peroxidation inhibitory effect [10]. Also, the peptides EEGER and GMEEER are potent antioxidants and were isolated from copra meal, a by-product of coconut oil [11].

Plant-derived peptides, especially from by-products like oilseed cakes, are gaining popularity due to the increased awareness of sustainable food systems [12,13]. Sacha Inchi (SI) (*Plukenetia volubilis*) is an oil-producing plant native to the Amazon region of South America that has significantly expanded into Africa and Asia due to its economic appeal [14]. The main by-product generated by the industry is the SI Oil Press-Cake (SIPC), which represents up to 50% of the raw seeds. Due to the quality of its proteins, SIPC has recently been used as a source of protein isolates and hydrolysates with techno-functional and bioactive properties [15,16,17]. However, the bioactive peptides responsible for the health benefits of SIPC may experience digestion, absorption, and transport from the gastrointestinal tract to tissues via the bloodstream, potentially resulting in structural modifications and subsequent changes in their biological effects [18]. Recently, protein hydrolysates have been obtained using digestive enzymes acting on SIPC, which exhibit antioxidant properties in in vitro assays and ex vivo studies [19,20]. Nonetheless, the identification of these sequences has been minimally investigated, and no research has yet examined the drug-likeness of the derived bioactive peptides.

Traditional methods for discovering bioactive peptides, involving substrate fractionation and enzymatic hydrolysis followed by purification and bioactivity assays, are laborious, time-consuming, and costly, often yielding inconsistent results [21]. Conversely, bioinformatic-driven in silico approaches offer a promising, cost-effective alternative for peptide discovery. Recently, an integrated methodology that combines aspects of both approaches has emerged, using peptides identified from food hydrolysates as a starting point for targeted bioinformatics analysis [22]. This study aims to advance research on the antioxidant peptides derived from SIPC by identifying potential candidates and assessing their bioactivity and bioavailability through a combination of in vitro, ex vivo, and in silico assays.

## 2. Materials and Methods

### 2.1. Materials

SIPC was kindly donated by SumaSach’a (Mosquera, Cundinamarca, Colombia), with a composition on the wet basis of 9.08 ± 0.03% of moisture, 54.47 ± 0.11% of protein, 6.84 ± 0.02% of fat, and 6.08 ± 0.02% of ash. All the chemicals (reagents and solvents) were of analytical grade and provided by Sigma-Aldrich (St. Louis, MO, USA), and Merck (Kenilworth, NJ, USA). For the absorbance and fluorescence measures, Sarstedt AG & Co. (Nümbrecht, Germany) and Corning Costar Corporation (Corning, NY, USA) supplied the 96-well and 48-well plates, respectively. An electrophoresis analysis was conducted using the equipment and reagents provided by Bio-Rad (Hercules, CA, USA). Biowest (Kansas City, MO, USA) supplied reagents and culture media for the assays conducted with RAW 264.7 macrophages, which were obtained from the American Type Culture Collection (ATCC) (Rockville, MD, USA).

### 2.2. Methods

#### 2.2.1. Protein Concentrate Production

SIPC adequacy (grinding, defatting, and sieving) and Sacha Inchi Protein Concentrate (SPC) production were conducted essentially using the procedure reported [23]. Briefly, defatted SIPC was mixed with deionized water (1:10 ratio, *w*/*v*) under alkaline conditions (pH 11.0), adjusted with NaOH in an Orion Star™ A211 pH meter (Thermo Fisher Scientific, Waltham, MA, USA), with continuous stirring (1 h, 800 rpm) at 60–70 °C. Upon centrifugation, the supernatant was retrieved, neutralized, and diafiltrated. Subsequently, the soluble protein solution was freeze-dried at 30 °C with a pressure of 0.08–0.1 mbar for 24 h in a bulk tray dryer/free zone console (Labconco, Kansas City, MO, USA). Finally, the freeze-dried soluble protein was stored at −18 °C until assays.

#### 2.2.2. Static Simulation of Gastro-Intestinal Digestion

The in vitro INFOGEST 2.0 method [24] was employed. Saliva gathered from twenty healthy volunteers was combined and preserved at −18 °C. Simulated gastric and intestinal fluids were prepared according to the protocol. Briefly, 3 g of SPC (n = 6) was dissolved in saliva at a ratio of 1:5 (*w*/*v*) and subjected to the oral phase for 5 min at 37 °C. The mixture was then diluted (1:1, *v*/*v*) with simulated gastric fluid (pH 3.0 adjusted with HCl) containing pepsin (EC 3.4.23.1) at an enzyme-to-substrate ratio (E:S) of 1:60 (*w*/*w*) and subjected to the gastric phase for 2 h. Samples (gastric digest, GD, n = 3) were collected after pepsin inactivation (pH adjustment to 7.0 with NaOH or HCl, heating at 95 °C for 15 min) and freezing at −40 °C. After completing the gastric phase, the samples were mixed with simulated intestinal fluid (1:1, *v*/*v*) containing pancreatin at an E:S ratio of 1:1.2 (*w*/*w*), and bile salts at a ratio of 1:30 (*w*/*w*) and subjected to the intestinal phase for 2 h. Samples (intestinal digest, ID, n = 3) were collected after pancreatin inactivation (heating at 95 °C for 15 min) following the same freezing protocol.

Subsequently, the samples were processed using the Sartorius^®^ ultrafiltration system (Vivaflow^®^ TFF Cassette, Göttingen, Germany). The digests underwent ultrafiltration with a 3 kDa membrane. The resulting permeate (containing molecules < 3 kDa) was collected as GD3 or ID3. The 3 kDa retentate was then filtered through a 10 kDa membrane, producing a permeate (>10 kDa) for GD1 or ID1, and a retentate (>3 kDa and <10 kDa) for GD2 or ID2. Blanks of digestion (labeled as B-plus the respective digest) without sample were also obtained and fractioned. All the samples were freeze-dried and stored at −18 °C until further analysis.

#### 2.2.3. Electrophoretic Profile

Sodium Dodecyl Sulfate–Polyacrylamide Gel Electrophoresis (SDS-PAGE) analysis was carried out to assess the protein profile of the SPC following the methodology reported [25]. A Criterion^TM^ cell system (Bio-Rad) was used. 75 µg of protein, quantified by using the Pierce bicinchoninic acid (BCA) kit (Thermo Fisher Scientific, Waltham, MA, USA), was loaded onto a 12% Bis-Tris Criterion™ XT Precast Gel polyacrylamide gel (Bio-Rad). Electrophoretic migration was performed initially at 100 V for 5 min, followed by 150 V for 1 h. The gel image was acquired using the Molecular Imager^®^ VersaDoc™ MP 4000 (Bio-Rad) and analyzed with Image Lab V6.1 software (Bio-Rad).

### 2.3. In Vitro Assays

The antioxidant activity was assessed using the ABTS and oxygen radical absorbance capacity (ORAC) assays, following the methods described [26]. For the ABTS assay, the absorbance was measured at 734 nm using a Biotek Synergy^TM^ HT plate reader (Winooski, VT, USA). For the ORAC assay, fluorescence readings were taken every 2 min over 120 min at excitation and emission wavelengths of 485 nm and 520 nm, respectively, using a FLUOstar Optima BMG Labtech plate reader controlled by FLUOstar Control V1.32 R2 software (Ortenberg, Germany). The Trolox equivalent antioxidant capacity (TEAC) and ORAC values were expressed as µmol Trolox equivalent (TE) per g of sample.

### 2.4. Ex Vivo Assays

#### 2.4.1. Macrophages Culture

RAW 264.7 mouse macrophages were cultured routinely in T75 flasks (Corning Costar Corporation) with modified Dulbecco’s Eagle medium (DMEM) supplemented with 10% fetal bovine serum (FBS) and 1% penicillin/streptomycin. The cells were maintained at 37 °C in a humidified atmosphere with 5% CO_2_ and 95% air. The culture medium was refreshed every 2–3 days. Upon reaching 80–90% confluence, the cells were harvested by gentle scraping, washed with phosphate-buffered saline (PBS) without Ca^2+^, Mg^2+^, or KCl, and centrifuged at 1000× *g* for 5 min to obtain cell pellets. The pellets were then resuspended in DMEM to prepare a stock solution. Cell counting was performed using an EVE™ Plus cell counter (NanoEntek, Seoul, Republic of Korea) to adjust the cell suspension density for subsequent assays.

#### 2.4.2. Cytotoxicity Assay

The 3-(4,5-dimethylthiazol-2-yl)-2,5-diphenyltetrazolium bromide (MTT) reduction assay, which measures the activity of cellular dehydrogenases, was determined according to the literature [27]. RAW 264.7 cells were seeded at 1 × 10^5^ cells/well in 96-well plates and incubated at 37 °C for 24 h. The medium was then replaced with 120 µL of sample (at various concentrations) in FBS-free DMEM, and the cells were further incubated for 24 h at 37 °C. The control treatment received FBS-free DMEM. After incubation, the cells were washed with PBS, treated with MTT solution (0.5 mg/mL) for 2 h at 37 °C, and the formazan crystals were dissolved in dimethyl sulfoxide (DMSO). The absorbance was measured at 570 nm using the Biotek Synergy^TM^ HT reader. The cell viability was calculated as a percentage relative to the control treatment, considered 100%.
(1)$$\mathrm{Cell}\;\mathrm{viability}\;(\%)=\left[\frac{Sample\;absorbance\;}{Control\;absorbance}\right]\cdot 100$$

#### 2.4.3. Reactive Oxygen Species (ROS) Production

The evaluation of the effect of the samples on the intracellular ROS levels was determined according to the literature [28]. The cells were seeded at 5 × 10^5^ cells/well in 48-well plates and incubated at 37 °C for 24 h. Then, the medium was discarded, and the cells were treated with 120 μL of sample (at two concentrations) dissolved in DMEM without FBS, and incubated for 24 h at 37 °C. Control cells received FBS-free DMEM, and the positive control cells received FBS-free DMEM supplemented with lipopolysaccharide (LPS, 10 µg/mL). After removing the treatment, 100 μL/well of 2′7′-dichlorofluorescein diacetate (DCFH-DA, 0.4 mg/mL), dissolved in Hank’s Balanced Salt Solution (HBSS), was added, and the plate was incubated for 1 h at 37 °C. The fluorescence was measured at 485 and 520 nm wavelengths of excitation and emission, respectively, by using the FLUOstar Optima BMG Labtech plate reader. The results were expressed as the % of control, which was considered to be 100%.

### 2.5. De Novo Peptides Sequencing

Analysis by liquid chromatography-tandem mass spectrometry (LC-MS/MS) was conducted by the Proteomics Facility at the Centro de Biología Molecular Severo Ochoa (CBM, CSIC-UAM, Madrid, Spain). An ion trap LTQ-Orbitrap-Velos-Pro hybrid mass spectrometer equipped with a nano-spray source and an Easy-nLC 1200 system (Thermo Scientific) was used, following the protocol outlined [25]. The peptidome was determined using mass spectrometric data, which was analyzed with the de novo PEAKS Studio V11.5 search engine (Bioinformatics Solutions Inc., Waterloo, ON, Canada). Peptides with an Average Local Confidence (ALC) ≥ 85%, calculated as the sum of the residue local confidence scores in the peptide divided by the peptide length, were selected for further analysis, ensuring high confidence in the peptide identifications.

### 2.6. In Silico Assays

#### 2.6.1. Resistant Peptides to Gastrointestinal Digestion

The data analysis was performed at the Biocomputational Analysis Core Facility (SABio, CBM). A comparison of the de novo peptide sequences of the ID sample and those of the SPC generated by PEAKS Studio V11.5 was performed using the find-pep-seq script [29], transforming peptide sequences into vectors. Based on the results, the cosine similarity measure (threshold of 0.95) was calculated [30]. This value ranged from 0 to 1, where 0 indicated that two vectors were orthogonal, and 1 indicated that they were identical in direction. Also, paired peptide sequences with a length difference of more than 8 amino acids were discarded. This ensured that the paired sequences were comparable in magnitude. Finally, the resulting ID sequences were deduplicated to ensure each entry was unique, and formatted into a FASTA file, which was used as the input in subsequent bioinformatics tools. Through this data analysis, the ID peptide sequences were considered as resistant peptides to in vitro gastrointestinal digestion.

#### 2.6.2. Antioxidant Properties and Bioavailability of Resistant Peptides

The resistant peptides were classified as antioxidants or non-antioxidants using the AnOxPP platform [8]. Subsequently, the AnOxPePred-1.0 tool [31] predicted the likelihood of the antioxidant peptides to scavenge free radicals, with the top 20 peptides selected for further in silico analysis: (a) the PlifePred server [32] was employed to predict the half-life of the peptides in blood and to estimate their hydrophobicity and hydrophilicity; (b) the peptide property calculator PepCalc was employed to estimate the molecular weight (MW), the isoelectric point (pI), and the length and solubility of the peptides [33]; (c) PASTA 2.0 was employed to estimate the secondary structure [34]; (d) ToxinPred [35] and AllerCatPro 2.0 [36] tools were employed to predict whether the selected resistant peptides were non-toxic or allergenic, respectively; (e) the SwissADME platform [37] was used to assess the bioavailability of the selected peptides, evaluating the drug-likeness parameters such as the Lipinski filter and the bioavailability score, pharmacokinetics, some physicochemical properties, and the lipophilicity; and (f) the probability of them being cell-penetrating peptides (CPP) was predicted using the MLCPP 2.0 tool [38].

#### 2.6.3. Cleavage Potential Sites of Antioxidant Peptides

The top 20 antioxidant-resistant peptides were subjected to protease-induced cleavage from the digestive system using the Rapid Peptides Generator (RPG) V2.0.3 (18 December 2023) software, with the command line “rpg -i input_file.fasta -o output_file.fasta -e 33 15 42 -d c,” where “-i” specifies the input .fasta file, “-o” designates the output file, and “-e 33, 15, 42 -d c” represents the pepsin, chymotrypsin, and trypsin enzymes in the concurrent mode from the RPG database used [39]. These enzymatic digestion parameters “pepsin (pH = 1.3) + chymotrypsin + trypsin” follow the literature recommendations, which simulate human gastrointestinal digestion [40].

### 2.7. Experimental Design and Statistical Analysis

The in vitro and ex vivo experiments were carried out under completely randomized designs, with each parameter evaluated independently and the measurements performed in triplicate at a minimum. An analysis of variance (ANOVA) and a test of significance (least significant differences (LSD) tests) were performed using SAS^®^ OnDemand Software (Cary, NC, USA, SAS Institute Inc., accessed in April 2024). The homogeneity of the variances was tested using Levene tests. The normality of the residuals was tested using a Shapiro–Wilks test. Differences were considered statistically significant at *p* < 0.05.

## 3. Results and Discussion

### 3.1. SDS-PAGE

SDS-PAGE under reducing conditions was conducted to analyze the impact of simulated gastrointestinal digestion and ultrafiltration on the alkaline soluble protein profile of SPC, illustrated in Figure 1. Lane (1), corresponding to the SPC sample, revealed fifteen protein bands ranging from 4.2 to 186.9 kDa. The use of alkaline water and moderate temperature was highly effective in extracting major proteins: albumins (◊), including the 3S albumin (27.5 kDa); globulins (*) at 47.4, 37.7, and 21.0 kDa; prolamins (▪) at 10.0, 6.3, and 4.2 kDa; and glutelins (+) at 69.9 kDa; all of them are well-documented in the literature [41,42,43]. Additionally, novel high MW polypeptides (¬) ranging from 186.9 to 86.8 kDa were observed, potentially including glutelins with enzymatic functions like beta-galactosidase, as suggested by gene ontology analysis, necessitating further investigation for validation.

In lines (2) and (3) (B-GD1 and B-ID1 fractions, respectively), the polypeptides (**꙳**) of 37.8 kDa, 63.7 kDa, and 35.6 kDa correspond to the enzymes utilized. The loss of these polypeptides was evident due to the ultrafiltration process, as shown in lines (6), (7), and (10), corresponding to the B-GD2, B-ID2, and B-ID3 fractions, respectively. In line (4) for GD1 fraction, the degradation of the higher MW polypeptides (¬), glutelins (+), and the globulin (*) (37.7 kDa), and the partial degradation of other proteins like the albumins (◊), globulins (*) (47.4, and 21.0 kDa), and prolamins (▪), initially present in SPC (line (1)), became evident once gastric digestion was completed. Similarly, line (5) for the ID1 fraction revealed that the intestinal digestion process was complete, yielding polypeptides (꙳) of 58.0, 33.0, 23.2, 16.6, and 11.3 kDa. Meanwhile, the fractionation process efficiently concentrated peptides (꙳) based on their molecular size: between 8.0 and 3.8 kDa for the GD2 fraction (line (8)); 3.1 kDa for the ID2 fraction (line (9)); and 3.7 kDa for the GD3 fraction (line (11)). Peptides were not detectable in the ID3 fraction (line (12)) due to the resolving gel pore size used.

These results were consistent with recent findings where the hydrolysis of glutelin from SI using pepsin and trypsin yielded peptides with MW < 3 kDa at 74% and 78%, respectively. The complete hydrolysis of all the fractions (albumin, globulin, and especially glutelin) was achieved sequentially, with trypsin proving more effective due to its greater number of cleavage sites. Authors have suggested that glutelins extracted via an alkaline solution may be more readily absorbed due to their rough surface with small pores, facilitating enzyme access to recognition sites. Additionally, the limited presence of antiparallel β-sheet structures in glutelins aids their hydrolysis [19]. These findings aligned closely with our previous results demonstrating the absence of intermolecular β-sheets in alkali-soluble proteins from SPC [23].

### 3.2. In Vitro Antioxidant Performance

To evaluate the effect of simulated digestion on the antioxidant activity of SPC, ABTS, and ORAC, assays were performed, as shown in Table 1. These assays utilized complementary mechanisms to mitigate the action of the free radicals. SPC exhibited significantly lower (*p* < 0.05) TEAC (30.85 µmol TE/g of sample) and ORAC (120.79 µmol TE/g of sample) values compared to its fractionated gastric and intestinal digests. However, the antioxidant capacity of the SPC was higher than that determined (TEAC and ORAC values of 0.49 and 0.11 µmol TE/g, respectively) in the ethanolic extract obtained from SI seeds [44].

In both assays, the antioxidant activity increased significantly after the action of the gastric and pancreatic enzymes. The highest antioxidant capacity was shown by GD2, followed by the ID3, and ID2 fractions. These results were consistent with Zhan and coworkers who reported ORAC values of 363.01, 313.62, and 264.74 µmol TE/g for the glutelin, albumin, and globulin fractions, respectively, representing more than double compared to the same non-digested Osborne protein fractions obtained from the SI meal [19]. Similarly, the TEAC values increased with the hydrolysis time and the sequential use of two enzymes compared to the single-step hydrolysis of SI protein concentrate, with values of 770, 950, 1100, 1190, and 1530 μmol TE/g after 240 min using Flavourzyme, Neutrase, Alcalase, Alcalase + Neutrase, and Alcalase + Flavourzyme, respectively [17]. Additionally, the antioxidant capacity of another protein concentrate derived from SI was 17.80 μmol TE/g, as assessed by the 2,2-diphenyl-1-picrylhydrazyl (DPPH) radical scavenging assay. However, this activity was significantly enhanced upon hydrolysis with Calotropis and crude papain, yielding values of 27.46 and 23.15 μmol TE/g, respectively [15]. These findings demonstrated that the hydrolysis process of the SIPC-proteins enhanced their antioxidant capacity by releasing bioactive peptides capable of counteracting oxidative agents.

### 3.3. Ex Vivo Antioxidant Capacity

#### 3.3.1. Cell Viability

The MTT assay was performed to evaluate the dose-dependent effect of SPC and its digest fractions on the viability of RAW264.7 macrophages and to select those non-toxic doses (>75% viability) for subsequent experiments. The doses ranged from 16 to 1000 μg sample/mL for the 24 h they were assayed. A significant dose-dependent reduction in the cell viability was observed for all the samples, except for the ID2 and ID3 fractions, as shown in Figure 2.

The doses of 500–1000, 1000, 1000, and 250–1000 μg/mL were found to be cytotoxic for the SPC, GD1, GD3, and ID1 fractions, respectively, resulting in a decrease in the cell viability higher than 25% compared to the control treatment. These findings were consistent with previous reports for an albumin fraction obtained from the SIPC, demonstrating that the cellular viability in the RAW264.7 macrophages significantly decreased when exposed to concentrations exceeding 320 μg/mL [45]. In contrast, the ID2 and ID3 fractions showed no cytotoxic effects at all the assayed doses, suggesting that the toxic compounds could be degraded during the action of the intestinal phase.

#### 3.3.2. Intracellular ROS Production in RAW264.7 Cells

To assess the ROS scavenging ability of the peptides in the SPC and their digest fractions, a DCFH-DA assay was conducted. The ROS production values of the RAW 264.7 cells under basal and stimulated conditions are shown in Figure 3. LPS treatment at 10 µg/mL significantly (*p* < 0.05) induced ROS production (122.12 ± 9.17%) in comparison to the control cells (100.00 ± 4.95%). These ROS-inducing effects had been previously reported [26]. In the case of the SPC sample (data not plotted), under basal conditions, treatment with 63 µg/mL did not produce any significant effect on the ROS levels (106.35 ± 10.43%, *p* > 0.05). However, the oxidative damage increased significantly at 125 µg/mL (118.57 ± 7.80, *p* < 0.05). Conversely, under the stimulated conditions, both 63 and 125 µg/mL of SPC significantly potentiated the LPS-inducing effects up to the ROS levels of 155.81 ± 9.42% and 131.20 ± 10.87%, respectively. This result indicated that the SPC induced oxidative damage in the macrophages, as evidenced by the high cytotoxic effects exerted by this sample (Section 3.3.1).

When the basal RAW264.7 macrophages were treated with two concentrations of each GD1, GD2, and GD3 fraction (Figure 3a, Figure 3c, and Figure 3e, respectively), a significant protective effect against oxidative stress was observed, leading to reduced ROS levels (*p* < 0.05) compared to the control treatment. The radical scavenging activity demonstrated in this study through the in vitro assays for these fractions could contribute to the observed ROS-reducing effects. Also, under the stimulated conditions, both fractions GD1 (16 µg/mL) and GD2 (500 and 1000 µg/mL) significantly reverted the inducing damage caused by the LPS (*p* < 0.05), while fraction GD3 at two assayed doses did not exert any effects. In the case of intestinal fractions, under basal conditions, the ID1 exerted significant ROS-inducing effects (*p* < 0.05), while the ID2 and ID3 did not exert any effects at both the doses of 500 and 1000 µg/mL (*p* > 0.05), compared with control. Finally, under the stimulated conditions, the ID1 at both doses and the ID3 at 500 µg/mL potentiated the ROS-inducing effects of the LPS (*p* < 0.05), whereas the ID2 and the highest dose of the ID3 showed the same effect (*p* > 0.05) as the LPS treatment alone (Figure 3b,d,f).

The antioxidant activity of the peptides could be attributed to their amino acid sequence, with the residues tyrosine, methionine, histidine, and phenylalanine being demonstrated to exert an important contribution [17]. Considering that pepsin cleaves the peptide bonds of the side chains of aromatic and hydrophobic amino acids such as leucine, phenylalanine, tryptophan, and tyrosine [39], significant free radical scavenging is attributed to the π-bonds in the structure of these amino acids, which act similarly to antioxidants such as flavonoids, polyphenols, and carotenoids, known for their conjugated π-systems in their molecular structure [7]. This could be the main reason for the protective effects exerted by gastric digest fractions in comparison to the intestinal digest fractions. This hypothesis is supported by evidence from a study in which pepsin was used to hydrolyze a byproduct derived from *Paralichthys olivaceus*, resulting in a pepsin protein hydrolysate that effectively mitigated ROS production in LPS-challenged RAW264.7 cells and zebrafish embryos compared to the hydrolysates obtained using other enzymes [46].

### 3.4. Peptidome Characterization

The peptidomes of the SPC and digest-derived fractions were characterized using the robust search engine PEAKS Studio. The key findings are summarized in Table 2. After gastric and intestinal digestion, a notable increase in the number of peptides was observed. Moreover, there was a notable rise in the percentage of peptides with medium chain lengths (6–10 amino acids (AA)) in the GD3, ID2, and ID3 fractions compared to the SPC. The proportion of short peptides (2–5 AA) remained consistent across the samples, with a slight decrease observed for the long peptides (>10 AA). This study pioneered the use of de novo sequencing to analyze the SI proteome. The ALC of 85% used as a benchmark ensured accurate peptide identification.

### 3.5. In Silico Analysis

#### 3.5.1. Antioxidant Potential of Resistant Peptides

Following LC-MS/MS analysis, the de novo peptides derived from the gastrointestinal digest (ID fraction) were compared to those from the SPC to identify the peptides resistant to the complete digestive process. A total of 416 peptides were identified with a cosine similarity value greater than 0.95 and a maximum sequence length difference of eight amino acids. The in vitro and ex vivo assay results demonstrated that both the gastric and intestinal digests exhibited high antioxidant activity. There is a particular interest in resistant peptides due to their potential bioavailability. In this context, the AnOxPP tool, employing artificial intelligence, predicted peptides with antioxidant properties based on a quantitative structure–activity relationship [8]. Out of the 416 peptides identified as resistant, 367 were classified as antioxidants, while the remaining 49 were classified as non-antioxidants. Then, the AnOxPePred-1.0 web server was utilized to assess the probability of peptides to scavenge free radicals (FRS), selecting the top 20 peptides for subsequent analysis as suggested in the literature [31].

The antioxidant probabilities of the resistant peptides, as well as their physicochemical characteristics evaluated using bioinformatics tools are presented in Table 3. The peptides SVMGPYYNSK and EWGGGGCGGGGGVSSLR exhibited the highest FR values, and the scores of the selected peptides ranged from 0.47 to 0.60. These findings were consistent with the values reported for the peptides initially identified in fish hydrolysates, and subsequently chemically synthesized and validated as antioxidants in in vitro assays [47]. The physicochemical characteristics of the peptides such as their hydrophobicity, attributed to amino acid residues like proline, leucine, valine, and alanine, enhanced their antioxidant activity by facilitating interactions with free radicals at the water–lipid interfaces. Conversely, hydrophilicity was provided by the amino acid residues, such as glycine promoting peptide flexibility and acting as hydrogen donors to neutralize the ROS [8]. 

The peptides RHWLPR and LQDWYDK exhibited high solubility in the water and contained significant proportions of aromatic amino acids (33.3% and 28.6%, respectively). These aromatic residues can stabilize ROS via electron or proton transfer, as previously reported [48].

Seventy-five percent of the peptides selected exhibited 100% coil structures, recently identified as predictive of bioactive peptides [48]. According to the predictions from ToxinPred and AllerTop, these peptides were assessed as non-toxic and non-allergenic. This property might be attributed to their length and secondary structure, particularly the random coil and extended strand conformations, known for their lower toxicity [49].

#### 3.5.2. Bioavailability Analysis of Antioxidant Peptides

Peptide bioavailability indicates effective utilization and intact systemic circulation post-oral ingestion [50]. Traditional drug development assesses absorption, distribution, metabolism, and excretion (ADME), with Lipinski’s rule-of-five (Ro5) setting thresholds for oral drug suitability (drug-likeness concept) based on physicochemical properties (MW, coefficient between n-octanol and water, water solubility, hydrogen donor, and acceptor) [51]. This tool predicts the peptide pharmacokinetics, including bioavailability, in early development phases. SwissADME uses machine learning to predict small molecule pharmacokinetics and drug-likeness [37]. Table 4 presents the bioavailability parameters for the selected peptides.

Eleven peptides lacked established parameters due to the size limitations (<200 characters) in Simplified Molecular Input Line Entry Specification (SMILES). Drug-likeness considers physicochemical traits like lipophilicity, assessed by the n-octanol/water partition coefficient (log *P*_o/w_) [37]. The log *P*_o/w_ values obtained for the selected peptides ranged between −2.2 and −10.14. These values classified the peptides as lipophilic, and they did not exceed the threshold established in Ro5: log *P*_o/w_ < 4.15. The parameter log *S* estimated the molecule’s water solubility; values more negative (<−10 per Ro5) indicated greater insolubility. All the selected peptides exhibited very high solubility. However, they did not meet the Ro5 thresholds for the MW (Table 3), the hydrogen bond acceptors (HBA), and the hydrogen bond donors (HBD), a MW between 150 and 500 g/mol, ≤10 HBA, and ≤5HBD, resulting in Lipinski violations.

The bioavailability score assessed the likelihood of the peptides achieving 10% oral bioavailability or measurable Caco-2 permeability based on factors like the total charge, the topological polar surface area, and the Lipinski filter violations [37]. The selected peptides had low scores in this parameter, ranging between 0.11 and 0.17. Finally, the predictions for the passive gastrointestinal absorption focused on the ability of a molecule to act as a substrate or inhibitor of the proteins that govern pharmacokinetic behaviors in humans. The selected peptides, however, showed a low classification in terms of their absorption potential. Although the antioxidant peptides did not meet the optimal bioavailability criteria for the small molecules under Ro5, their bioactivity was consistent with the findings reported in the literature. Given the inherent challenges in the peptide drug development—such as membrane impermeability and limited in vitro stability—it is estimated that only about 10% of the studied peptides overcame these barriers to achieve bioavailability, while the remaining 90% typically targeted extracellular receptors [52]. In our study, the majority of the selected and analyzed peptides encountered these challenges, as shown in Table 4. These peptides might bind to cell surface receptors, initiating specific intracellular effects similar to proteins and antibodies, potentially benefiting gastrointestinal tissues [53]. However, these studies need further validation.

The peptides SVMGPYYNSK, RHWLPR, SGLSGGGYGSNK, and FSSSSGYGGSSR showed potential as cell-penetrating peptides that could facilitate the transport of high-MW polar molecules as “Trojan horses” [54]. These peptides have a blood half-life averaging 816 s (13.6 min), similar to other bioactive peptides and circulating hormones [48]. These functional characteristics complement the antioxidant potential of these peptides.

#### 3.5.3. Cleavage Modification Analysis

Miscleavage is a phenomenon where an enzyme fails to cleave at its preferred site, potentially due to biological, chemical, or physical factors [39]. Therefore, the top 20 antioxidant peptides, identified as resistant under in vitro gastrointestinal digestion (Section 3.5.1), were subjected to an additional in silico (without miscleavages) hydrolysis to determine potential cleavage sites, as well as the possible alteration of their antioxidant activity and bioavailability, using the platforms AnOxPePred-1.0 and Swis-sADME, respectively. The results are shown in Appendix A.

Forty-six peptides were obtained, with lengths ranging from 2 to 13 amino acids (AA). Regarding antioxidant capacity, twelve of these peptides exhibited high probabilities (≥0.50) of being FRS. Among them, the sequences SVMGPY, EW, GGGGCGGGGGVSS, PQY, HGGGGGG, GGGG, HW, and SGGGY came from the resistant peptides with a high probability of being FRS (Table 3), while VVTAYPER and EY were released from the other FRS peptides. In several of the top 20 antioxidant peptides and the 46 fragments obtained (Appendix A), the presence of the SG sequence in their structure was important, because it had been demonstrated that the peptide SGP, obtained from the soy proteins (*Glycine max*), effectively scavenges superoxide radicals and protects HepG2 cells from H_2_O_2_-induced oxidative damage [40].

Regarding bioavailability, the average number of Lipinski violations and the bioavailability score according to the Ro5 improved from 3 and 0.17 (Table 4) to 1.3 and 0.33, respectively, for this group of peptides. This indicates that the cleavage process enhanced the physicochemical properties of the top 20 antioxidant peptides, making them promising candidates for effective drug development. These peptides demonstrated an increased capacity for absorption and systemic circulation in the human body [51], without having toxic or allergenic effects, as evaluated using the tools ToxinPred and AllerCatPro 2.0, respectively.

## 4. Conclusions

The alkali-soluble proteins from the SPC underwent simulated gastrointestinal digestion using the INFOGEST 2.0 protocol. In vitro assays, including ABTS and ORAC, revealed that hydrolysis allowed the release of bioactive peptides with strong antioxidant properties. These effects were corroborated in ex vivo assays, where SPC digests reduced the ROS levels and mitigated dysfunction in the RAW264.7 cells. This study marked the first use of de novo sequencing to assess the SI peptidome. Despite the low bioavailability indices predicted by in silico tools, the selected resistant peptides showed promising potential for extracellular targets and drug delivery systems. Similarly, under ideal gastrointestinal digestion conditions, it has been shown that this group of peptides can generate smaller fragments with high antioxidant and potential bioavailability. However, this must be corroborated through future research, which should focus on the chemical synthesis of identified sequences—SVMGPYYNSK, EWGGGGCGGGGGVSSLR, RHWLPR, LQDWYDK, and ALEETNYELEK—to validate their antioxidant activity and bioavailability in experimental and in vivo models.

## Figures and Tables

**Figure 1 foods-13-03924-f001:**
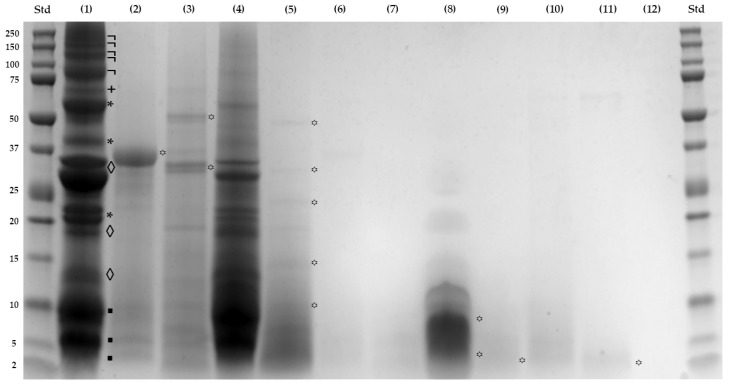
Electrophoretic (SDS-PAGE) analysis. (Std) Lines with Precision Plus Protein^TM^ Dual Xtra Standard. (1) Sacha Inchi Protein Concentrate (SPC); (2) gastric (B-GD1) and (3) intestinal (B-ID1) (Blanks, >10 kDa); (4) gastric (GD1) and (5) intestinal (ID1) (digests, >10 kDa); (6) gastric (B-GD2) and (7) intestinal (B-ID2) (Blanks, 3–10 kDa); (8) gastric (GD2) and (9) intestinal (ID2) (digests, 3–10 kDa); (10) intestinal (B-ID3) (Blank, <3 kDa); (11) gastric (GD3) and (12) intestinal (ID3) (digests, <3 kDa). Note the different polypeptides in the lines classified in the albumins (◊), globulins (*), prolamins (▪), glutelins (+), and higher molecular weight polypeptides (¬), as well as the various peptides derived from the digestion assay (꙳).

**Figure 2 foods-13-03924-f002:**
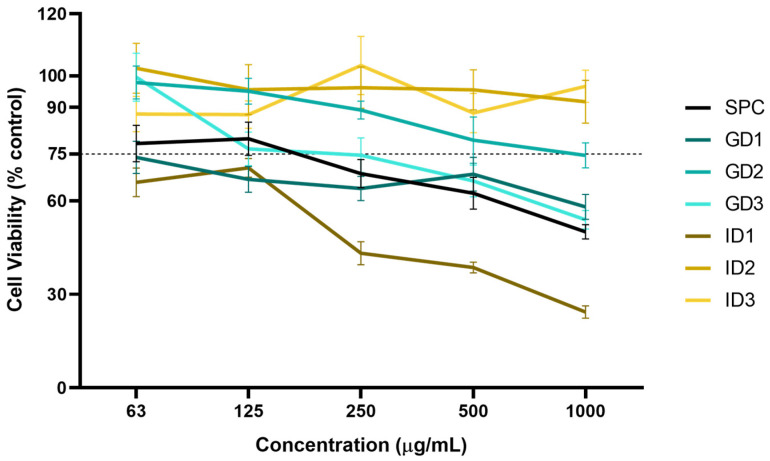
Viability of RAW 264.7 cells treated with Sacha Inchi Protein Concentrate and its fractioned digests. Control treatment represents 100.00 ± 3.83% cell viability. Doses below the dashed black line (75%) are toxic. SPC: Sacha Inchi Protein Concentrate; GD1 and ID1: gastric and intestinal digests fractions (>10 kDa), respectively; GD2 and ID2: gastric and intestinal digests fractions (3–10 kDa), respectively; GD3 and ID3: gastric and intestinal digests fractions (<3 kDa), respectively. Values are presented as mean ± SD, n = 8.

**Figure 3 foods-13-03924-f003:**
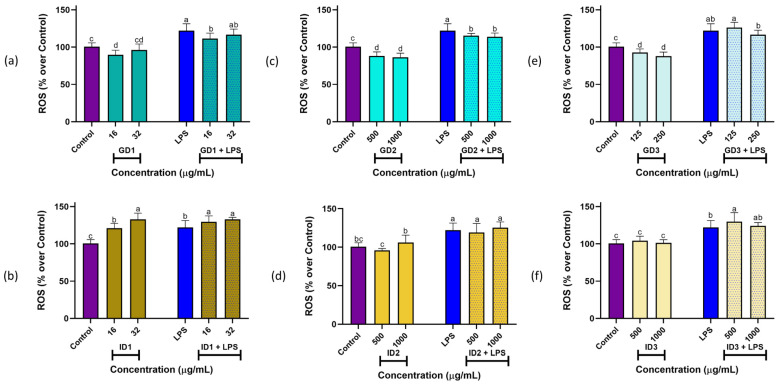
Reactive oxygen species (ROS) production (expressed as %) in RAW264.7 macrophages under basal and stimulated conditions. (**a**) Gastric (GD1) and (**b**) intestinal (ID1) (digests, >10 kDa); (**c**) gastric (GD2) and (**d**) intestinal (ID2) (digests, 3–10 kDa); (**e**) gastric (GD3) and (**f**) intestinal (ID3) (digests, <3 kDa). Lipopolysaccharide (LPS) stimulus (10 μg/well). ^a–d^ Different letters indicate significant differences between samples (LSD Test, *p* < 0.05).

**Table 1 foods-13-03924-t001:** TEAC and ORAC assays of Sacha Inchi Protein Concentrate (SPC) after simulated gastro-intestinal digestion and ultrafiltration.

Sample	TEAC (µmol TE/g)	ORAC (µmol TE/g)
SPC	30.85 ^f^ ± 0.83	120.79 ^f^ ± 6.34
GD1	131.23 ^e^ ± 9.50	284.36 ^e^ ± 25.31
GD2	577.84 ^a^ ± 20.69	719.47 ^a^ ± 59.69
GD3	139.53 ^de^ ± 11.11	268.44 ^e^ ± 18.96
ID1	158.30 ^d^ ± 7.34	398.67 ^d^ ± 30.98
ID2	188.56 ^c^ ± 18.21	490.14 ^c^ ± 46.98
ID3	325.60 ^b^ ± 22.87	669.04 ^b^ ± 60.59

^a–f^ Different letters indicate significant differences among the values of the same column (LSD test, *p* < 0.05). Values are expressed as the µmol equivalent of the Trolox (TE)/g of the sample (mean ± SD, n = 8). SPC: Sacha Inchi Protein Concentrate; GD1 and ID1: gastric and intestinal digests fractions (>10 kDa), respectively; GD2 and ID2: gastric and intestinal digests fractions (3–10 kDa), respectively; GD3 and ID3: gastric and intestinal digests fractions (<3 kDa), respectively.

**Table 2 foods-13-03924-t002:** Summary of de novo peptidome characterization of Sacha Inchi Protein Concentrate (SPC) after simulated gastro-intestinal digestion and ultrafiltration.

Sample	Number of Peptides ^a^	Peptide Chain Length ^b^ (%)		
		Short (2–5 AA)	Medium (6–10 AA)	Long (>10 AA)
SPC	1819	0.00	56.18	43.82
GD	2862	0.03	61.29	38.68
GD1	1889	0.05	59.93	40.02
GD2	1248	0.00	67.23	32.77
GD3	937	0.11	72.68	27.21
ID	4095	0.02	62.78	37.19
ID1	2821	0.04	59.09	40.87
ID2	1264	0.00	71.91	28.09
ID3	1069	0.00	76.43	23.57

SPC: Sacha Inchi Protein Concentrate; GD and ID: gastric and intestinal digests; GD1 and ID1: gastric and intestinal digests fractions (>10 kDa), respectively; GD2 and ID2: gastric and intestinal digests fractions (3–10 kDa), respectively; GD3 and ID3: gastric and intestinal digests fractions (<3 kDa), respectively. ^a^ De novo peptides with Average Local Confidence (ALC) ≥ 85%. ^b^ Classified by the number of amino acid (AA) residues.

**Table 3 foods-13-03924-t003:** Properties of antioxidant peptides identified in a Sacha Inchi Protein Concentrate (SPC) using in silico methods.

Sequence	AnOxPePred-1.0 ^a^	PlifePred ^b^		PepCalc ^c^				Pasta 2.0 ^d^		
								Secondary Structure		
	FRS	Hydrophob.	Hydrophil.	MW	pI	Length	Sol.	α-Helix	β-Strand	coil
SVMGPYYNSK	0.60	−0.13	−0.36	1145.43	9.34	10	Poor	0.00	0.00	100.00
EWGGGGCGGGGGVSSLR	0.58	0.00	−0.06	1492.58	6.14	17	Poor	0.00	0.00	100.00
VALLPQYVDPK	0.54	−0.02	−0.29	1242.46	6.55	11	Poor	0.00	0.00	100.00
HGGGGGGFGGGGFSR	0.54	0.03	−0.15	1263.28	10.59	15	Poor	0.00	0.00	100.00
FSSSSGYGGGSSR	0.52	−0.16	0.00	1235.22	9.59	13	Good	0.00	0.00	100.00
RHWLPR	0.52	−0.52	0.05	864.01	12.01	6	Good	0.00	0.00	100.00
SGLSGGGYGSNK	0.51	−0.10	0.00	1083.11	9.50	12	Good	0.00	0.00	100.00
HGGGGGGFGGGGFDK	0.51	0.04	0.03	1263.28	7.56	15	Good	0.00	0.00	100.00
VTGGGFGGSR	0.49	−0.03	−0.11	893.94	10.81	10	Poor	0.00	0.00	100.00
ALEETNYELEK	0.49	−0.28	0.76	1338.42	3.67	11	Good	18.18	0.00	81.82
LVGPDGPDKMVK	0.49	−0.13	0.49	1255.49	6.76	12	Good	0.00	0.00	100.00
VMLYNCK	0.48	−0.05	−0.67	870.10	8.75	7	Poor	0.00	28.57	71.43
LVGPNGLCMDVK	0.48	0.02	−0.22	1245.52	5.92	12	Poor	0.00	0.00	100.00
LQDWYDK	0.48	−0.33	0.24	967.03	3.71	7	Good	0.00	0.00	100.00
SGGFGGNFGNR	0.48	−0.12	−0.12	1069.09	10.57	11	Poor	0.00	0.00	100.00
ALEESNYELEK	0.48	−0.29	0.83	1324.39	3.67	11	Good	0.00	0.00	100.00
YVVTAYPER	0.48	−0.14	−0.28	1097.22	6.56	9	Good	0.00	33.33	66.67
VVDNFFNDFLPR	0.48	−0.09	−0.24	1482.64	3.71	12	Good	33.33	0.00	66.67
LTSEGFEYVNMK	0.47	−0.11	−0.02	1417.58	4.15	12	Good	0.00	16.67	83.33
FSSSSGYGGSSR	0.47	−0.18	0.00	1178.17	9.59	12	Good	0.00	0.00	100.00

^a^ Probabilities of free radical scavenging (FRS) were calculated using AnOxPePred-1.0 (https://services.healthtech.dtu.dk/service.php?AnOxPePred-1.0, accessed on 2 December 2024). ^b^ Hydrophobicity and hydrophilicity were calculated using PlifePred (https://webs.iiitd.edu.in/raghava/plifepred/, accessed on 2 December 2024). ^c^ Molecular weight (MW), and solubility (Sol.) based on the combined results of isoelectric point (pI) and length were estimated using PepCalc (http://pepcalc.com/, accessed on 2 December 2024). ^d^ Portions of secondary structure were estimated using Pasta 2.0 (http://old.protein.bio.unipd.it/pasta2/, accessed on 2 December 2024). Conventions: A, Alanine; C, Cysteine; D, Aspartic acid; E, Glutamic acid; F, Phenylalanine; G, Glycine; H, Histidine; K, Lysine; L, Leucine; M, Methionine; N, Asparagine; P, Proline; Q, Glutamine; R, Arginine; S, Serine; T, Threonine; V, Valine; W, Tryptophan; Y, Tyrosine. Tools were accessed in April and May 2024.

**Table 4 foods-13-03924-t004:** Bioavailability of antioxidant peptides identified in a Sacha Inchi Protein Concentrate (SPC) using bioinformatic tools.

Peptide	Lipophilicity ^a^	Properties ^a^			Drug-Likeness ^a^		Pharmacokinetics ^a^	CPP ^b^		PlifePred ^c^
	Log Po/w	Log S	HBA	HBD	LVN	BS	GI Absorp.	Class.	Prob.	Half-Life
SVMGPYYNSK	−5.16	−4.52	18	16	3	0.17	Low	CPP	0.74	817.61
EWGGGGCGGGGGVSSLR	-	-	-	-	-	-	-	Non-CPP	0.12	842.71
VALLPQYVDPK	-	-	-	-	-	-	-	Non-CPP	0.09	795.01
HGGGGGGFGGGGFSR	−9.49	−1.92	20	21	3	0.17	Low	Non-CPP	0.40	834.71
FSSSSGYGGGSSR	-	-	-	-	-	-	-	Non-CPP	0.12	825.61
RHWLPR	−2.8	−4.38	11	14	3	0.17	Low	CPP	0.72	839.91
SGLSGGGYGSNK	−10.14	−0.23	20	19	3	0.17	Low	CPP	0.75	807.21
HGGGGGGFGGGGFDK	−11	0.05	21	19	3	0.17	Low	Non-CPP	0.39	834.81
VTGGGFGGSR	−8.04	−0.19	15	16	3	0.17	Low	Non-CPP	0.31	839.91
ALEETNYELEK	-	-	-	-	-	-	-	Non-CPP	0.14	887.71
LVGPDGPDKMVK	-	-	-	-	-	-	-	Non-CPP	0.44	905.41
VMLYNCK	−2.19	−5.5	12	11	3	0.17	Low	Non-CPP	0.11	763.11
LVGPNGLCMDVK	-	-	-	-	-	-	-	Non-CPP	0.38	717.01
LQDWYDK	−6.49	−1.59	16	14	3	0.11	Low	Non-CPP	0.18	871.61
SGGFGGNFGNR	−9.27	−0.91	17	18	3	0.17	Low	Non-CPP	0.33	830.11
ALEESNYELEK	-	-	-	-	-	-	-	Non-CPP	0.22	926.01
YVVTAYPER	-	-	-	-	-	-	-	Non-CPP	0.09	759.41
VVDNFFNDFLPR	-	-	-	-	-	-	-	Non-CPP	0.50	839.01
LTSEGFEYVNMK	-	-	-	-	-	-	-	Non-CPP	0.08	555.41
FSSSSGYGGSSR	-	-	-	-	-	-	-	CPP	0.88	832.01

^a^ The bioavailability was evaluated using SwissADME (http://www.swissadme.ch/): the lipophilicity was evaluated by the partition coefficient between n-octanol and water (log *P*_o/w_); the physicochemical properties include the solubility parameter (log *S*), the number of hydrogen bond acceptors (HBA), and the number of hydrogen bond donors (HBD); drug-likeness includes the Lipinski violations number (LVN), and the bioavailability score (BS); pharmacokinetics includes the classification for gastro-intestinal absorption (GI Absorp.). ^b^ MLCPP 2.0 was used to estimate the probability of being cell-penetrating peptides (CPP) (https://balalab-skku.org/mlcpp2/). ^c^ PlifePred was employed to predict the half-life in blood (https://webs.iiitd.edu.in/raghava/plifepred/). Conventions: A, Alanine; C, Cysteine; D, Aspartic acid; E, Glutamic acid; F, Phenylalanine; G, Glycine; H, Histidine; K, Lysine; L, Leucine; M, Methionine; N, Asparagine; P, Proline; Q, Glutamine; R, Arginine; S, Serine; T, Threonine; V, Valine; W, Tryptophan; Y, Tyrosine. The tools were accessed in November 2024.

## Data Availability

The data presented in this study are available on request from the corresponding author due to restrictions on the confidential information that they are immersed in.

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
