# Peer review of "Antioxidant Peptides from Sacha Inchi Meal: An In Vitro, Ex Vivo, and In Silico Approach"

_foods, 2024, doi:10.3390/foods13233924_

Round 1
Reviewer 1 Report
Comments and Suggestions for Authors
The manuscript titled “Antioxidant Peptides from Sacha Inchi Meal: An In vitro, Ex vivo, and In silico Approach” (Manuscript number: foods-3302397) used the protein concentrate (SPC) from Sacha Inchi Oil Press-Cake (SIPC) to prepare antioxidant peptides through a combination of in-vitro, ex-vivo and in-silico methods. It suggested the released peptides from SPC during stimulated gastrointestinal digestion had higher ABTS and ORAC values, and especially the gastric digest fractions GD1, GD2, and GD3 (<3, 3-10, and 23 >10 kDa, respectively), separated by ultrafiltration, significantly reduced ROS levels in RAW264.7 cell lines. Moreover, authors used bioinformatic tools characterized 315 antioxidant peptides with resistant digestion. This work provides valuable data for the comprehensive utilization of SIPC, a by-product of oil processing from Sacha Inchi fruits. However, several points in the manuscript have to be addressed. Specific issues are listed as follows.
1/For the section Keywords, “resistant peptides” should be changed “antioxidant peptides”.
2/Line 72-74, References are necessary.
3/Line 83, the basic composition for SIPC donated by SumaSach’a should be indicated.
4/Line 90, the source of RAW 264.7 macrophages should be indicated.
5/Line 100-101, the approved document for the “twenty healthy volunteers” should be added.
6/In the section Methods, the method adjusting the samples to pH11.0, pH3.0, and pH7.0 (Line 95, Line 104, and Line 107) should be presented.
7/Line 144, parameters for PBS should be indicated.
8/Line 110-111, (1) 2 h is enough time for the intestinal phase of SPC digestion? (2) how to inactivate pancreatin?
9/Line 128, producer information for Bio-Rad should be indicated.
10/Line 147, what means “the desired density”? Should be cleared.
11/Line 158, produce information for “Biotek SynergyTM HT reader” should be indicated.
12/Line 202, “PepCalc” is software?
13/Line 212, “A The in vitro and ex vivo experiments” is right?
14/Line 221, “T SDS-PAGE” is right?
15/Line 246, could author present a better figure of SDS-PAGE for this manuscript?
16/Line 255-257, Which reference showing the recent findings?
17/Line 266, “T To evaluate the effect…”is right? Please check in the errors existing in the whole manuscript.
18/Line 273, “TEAC” should be defined when it appeared firstly (see table 1).
19/Line 290-295, (1) “SI protein concentrate” indicates what? (2) Not clear logic connection between them.
20/Line 267-272, lines 304-306 and Lines 364-367, background information for the methods is not introduced again.
21/Line 317 for Figure 2 “Viability of RAW 264.7 cells treated with Sacha Inchi Protein Concentrate and its fractioned digests”, this figure should be redrawn because of strange marked values of the coordinate system, and where is the curve of the control group?
22/Lines 287-289, what means “These results were consistent with Zhan and coworkers that reported ORAC values of 363.01, 313.62, and 264.74 μmol TE/g for the glutelin, albumin, and globulin fractions, respectively”? what the comparison makes sense? Should be cleared.
23/Lines 301-315, the results need to be discussed fully.
24/Line 344-346, Unlike ID1, why ID2 and ID3 fractions at both doses of 500 and 1000 μg/mL did not exert any effect on the LPS-inducing ROS? Should be discussed.
25/Line 423-425, what sense for this work?
26/Line 433-435, “However, they did not meet Ro5 thresholds for MW (Table 3), hydrogen bond acceptors (HBA), and hydrogen bond donors (HBD): MW between 150 and 500 g/mol, ≤ 10 HBA, and ≤ 5 HBD, resulting in Lipinski violations”. So, why authors select Lipinski's of-five (Ro5) for this work, as indicated from line 419-421?
27/Line 470-471, what means “the selected peptides exhibited a low classification”? Not clear.
28/Line 472-483, (1) Actually, a big difference between the predictive values and “real” test values in the resistant peptides identified as antioxidant properties often take place, and thus the activity of the resistant peptides predicted via in-silico tools should be confirmed using man-made synthesized peptides. It is important for improving the novelty of this work. (2) Line 472-473, where to confirm this point? (3) Line 479-483, “the peptides RHWLPR and ALEETNYELEK showed potential as cell-penetrating peptides, facilitating the transport of high MW polar molecules as 'Trojan horses' [33]. Another critical parameter is the peptides' half-life in blood, averaging 816 seconds (13.6 minutes), that is consistent with other bioactive peptides and circulating hormones [47]”. The predicated functionality of the two peptides is linked to their antioxidant activity directly?
29/Line 488-489, “alleviated stress” indicates which “stress”? Not clear.
30/Section “3.1. In vitro antioxidant performance” should be changed to section “3.2. In vitro antioxidant performance”.
31/Line 501, National name for funding this grant should be supplemented.
32/For references, line 596 “Plukenetia Volubilis L.” should be written in italic.
Author Response
Response to Reviewer 1
Thank you very much for taking the time to review this manuscript, and especially for your depth analysis and detail, which has allowed us to consider and discuss many aspects we had not previously addressed. Your contributions have been very valuable in improving our manuscript. Please find the detailed responses below and the corresponding revisions/corrections highlighted in track changes in the re-submitted file.
- For the section Keywords, “resistant peptides” should be changed “antioxidant peptides”.
Answer: As suggested, the keyword has been changed (line 35).
- Line 72-74, References are necessary.
Answer: As suggested, a reference was added (line 86).
- Line 83, the basic composition for SIPC donated by SumaSach’a should be indicated.
Answer: The basic proximal composition of SIPC has been added as suggested (lines 97-98).
- Line 90, the source of RAW 264.7 macrophages should be indicated.
Answer: The source of the RAW 264.7 macrophages has been specified in the materials section (lines 104–105).
- Line 100-101, the approved document for the “twenty healthy volunteers” should be added.
Answer: Thank you for pointing this out. We have included the Informed Consent Statement (lines 523-529) in accordance with the MDPI guidelines, specifically the Research Involving Human Subjects section of the Research and Publication Ethics, to clarify this matter.
- In the section Methods, the method adjusting the samples to pH11.0, pH3.0, and pH7.0 (Line 95, Line 104, and Line 107) should be presented.
Answer: As suggested, the pH adjustment method was added (lines 111-112, 122, and 125, respectively).
- Line 144, parameters for PBS should be indicated.
Answer: As suggested, the PBS parameters have been included (lines 163-164).
- Line 110-111, (1) 2 h is enough time for the intestinal phase of SPC digestion? (2) how to inactivate pancreatin?
Answer: According to the INFOGEST 2.0 protocol, the intestinal phase has been estimated in 2 hours. (2) The pancreatin inactivation method was included as per the cited protocol (line 129-130).
- Line 128, producer information for Bio-Rad should be indicated.
Answer: This information is specified in the Materials section (lines 103).
- Line 147, what means “the desired density”? Should be cleared.
Answer: The word “desired” has been deleted (line 167).
- Line 158, produce information for “Biotek SynergyTM HT reader” should be indicated.
Answer: This information has been added (line 151). Additionally, the acronym 'TM' was superscripted.
- Line 202, “PepCalc” is software?
Answer: PepCalc is an online bioinformatics calculation tool designed to analyze the physicochemical properties of peptides. The other tools used for in silico analyses are also online bioinformatics tools. For clarity regarding each tool, this paragraph was modified (lines 218–231).
- Line 212, “A The in vitro and ex vivo experiments” is right?
Answer: It is incorrect. Thank you for pointing out the error; we have corrected it (line 233).
- Line 221, “T SDS-PAGE” is right?
Answer: It is incorrect. Thank you for pointing out the error; we have corrected it (line 242).
- Line 246, could author present a better figure of SDS-PAGE for this manuscript?
Answer: It is not possible to provide a better figure; however, we have slightly adjusted the image by increasing its brightness by 20% to improve the resolution of the bands.
- Line 255-257, Which reference showing the recent findings?
Answer: Thank you for your observation. Previous studies on SI proteins have identified these bands, as mentioned in line 248.
- Line 266, “T To evaluate the effect…”is right? Please check in the errors existing in the whole manuscript.
Answer: It is incorrect. Thank you for pointing out the errors; we have corrected them (line 286).
- Line 273, “TEAC” should be defined when it appeared firstly (see table 1).
Answer: TEAC has been defined as suggested (lines 155).
- Line 290-295, (1) “SI protein concentrate” indicates what? (2) Not clear logic connection between them.
Answer: Thank you for pointing this out. (1) In other studies conducted on SI, protein concentrates have been obtained, and their antioxidant capacity has been measured. (2) Research findings on SI are described, demonstrating that the hydrolysis process enhances the antioxidant capacity of the resulting hydrolysates compared with initial protein concentrate. This paragraph has been rephrased for greater clarity (lines 304-313).
- Line 267-272, lines 304-306 and Lines 364-367, background information for the methods is not introduced again.
Answer: Thank you for bringing this to our attention. The background information regarding the methods has been deleted.
- Line 317 for Figure 2 “Viability of RAW 264.7 cells treated with Sacha Inchi Protein Concentrate and its fractioned digests”, this figure should be redrawn because of strange marked values of the coordinate system, and where is the curve of the control group?
Answer: Thank you for pointing this out. We have smoothed the curves, adjusted the y-axis control value, and added a clarifying note in the caption (line 325).
- Lines 287-289, what means “These results were consistent with Zhan and coworkers that reported ORAC values of 363.01, 313.62, and 264.74 μmol TE/g for the glutelin, albumin, and globulin fractions, respectively”? what the comparison makes sense? Should be cleared.
Answer: This refers to protein fractions obtained from the SI meal, for which the authors assessed antioxidant capacity. Our comparison aims to relate our research to that of the cited authors. We have clarified the paragraph (lines 304-305).
- Lines 301-315, the results need to be discussed fully.
Answer: A small explanation of probable reasons of the lack of toxicity of the intestinal digests has been added (lines 335-337).
- Line 344-346, Unlike ID1, why ID2 and ID3 fractions at both doses of 500 and 1000 μg/mL did not exert any effect on the LPS-inducing ROS? Should be discussed.
Answer: Thank you for bringing this to our attention. We have expanded the discussion for greater clarity (lines 377-382).
- Line 423-425, what sense for this work?
Answer: Thank you for pointing this out. As suggested, we have better described the purpose of using ADME evaluation in peptides (line 452-453).
- Line 433-435, “However, they did not meet Ro5 thresholds for MW (Table 3), hydrogen bond acceptors (HBA), and hydrogen bond donors (HBD): MW between 150 and 500 g/mol, ≤ 10 HBA, and ≤ 5 HBD, resulting in Lipinski violations”. So, why authors select Lipinski's of-five (Ro5) for this work, as indicated from line 419-421?
Answer: The authors selected Lipinski's Rule of Five (Ro5) because it is a standard method for assessing drug-likeness compounds.
- Line 470-471, what means “the selected peptides exhibited a low classification”? Not clear.
Answer: Thank you for pointing this out. We have corrected the sentence to improve the description of the data (486-487).
- Line 472-483, (1) Actually, a big difference between the predictive values and “real” test values in the resistant peptides identified as antioxidant properties often take place, and thus the activity of the resistant peptides predicted via in-silico tools should be confirmed using man-made synthesized peptides. It is important for improving the novelty of this work. (2) Line 472-473, where to confirm this point? (3) Line 479-483, “the peptides RHWLPR and ALEETNYELEK showed potential as cell-penetrating peptides, facilitating the transport of high MW polar molecules as 'Trojan horses' [33]. Another critical parameter is the peptides' half-life in blood, averaging 816 seconds (13.6 minutes), that is consistent with other bioactive peptides and circulating hormones [47]”. The predicated functionality of the two peptides is linked to their antioxidant activity directly?
Answer: (1) We understand that this is a limitation of our study, and we agree that additional studies are necessary. (2) We have included this limitation (lines 511-514). (3) These are complementary characteristics of antioxidant peptides. We have made this clarification (line 502).
- Line 488-489, “alleviated stress” indicates which “stress”? Not clear.
Answer: Thank you for pointing this out. This sentence has been deleted for better clarity.
- Section “3.1. In vitro antioxidant performance” should be changed to section “3.2. In vitro antioxidant performance”.
Answer: Thank you for bringing this to our attention. We have corrected the numbering.
- Line 501, National name for funding this grant should be supplemented.
Answer: Thank you for pointing this out. We have removed this funding institution from the acknowledgments section and have added the national name for the funding source in the funding section.
- For references, line 596 “Plukenetia Volubilis L.” should be written in italic.
Answer: Thank you for your observation. This modification in the bibliography will be made later during the final editing stage if the article is accepted for publication.
Reviewer 2 Report
Comments and Suggestions for Authors
The manuscript seems interesting and needs revision.

Comments on the Quality of English Language
English language needs to be more concise.
Author Response
Response to Reviewer 2 Comments
Thank you very much for taking the time to review this manuscript. Your detailed comments have helped us improve the document. Please find the detailed responses below and the corresponding revisions/corrections highlighted/in track changes in the re-submitted file.
- Line 30-32: I think this information is not required.
Answer: As suggested, the sentence has been deleted in the abstract.
- Line 17: protein concentrate (SPC)……what does ‘S’ means?
Answer: Thank you for bringing this point to our attention. The acronym SPC refers to the Sacha Inchi Protein Concentrate obtained. We have added this information (line 18) for clarification.
- Line 43: and these, into harmless??? Please revise.
Answer: The sentence has been corrected (lines 44-45).
- Line 212 and 221: Revise the sentence.
Answer: As suggested, both sentences have been revised (lines 234 and 243).
- Line 258: Provide the reference.
Answer: The reference has been added (line 283).
- Line 266: Why is the ‘T To evaluate…’.
Answer: It is incorrect. Thank you for pointing out the error; we have corrected it (line 287).
- Line 273: What is TEAC?
Answer: TEAC has been defined (line 155).
- Line 364-367: I think this information is not required.
Answer: The information has been deleted as suggested.
- Please check the grammatical errors throughout the manuscript.
Answer: The manuscript has been revised and corrected accordingly.
- The language of the manuscript needs to be more concise.
Answer: As suggested, the language has been revised and modified.
- Well, the idea of manuscript seems interesting. The manuscript to be more concise.
Answer: We understand that our study has limitations, and we agree that additional studies should be necessary. We have stated this limitation in the conclusions section.
- Is there any commercialized production of antioxidant peptides from Sacha Inchi Meal currently?
Answer: To the best of our knowledge, no products have been commercialized yet.
Reviewer 3 Report
Comments and Suggestions for Authors
The aim of using bioinformatics to create functional peptides from food waste is fascinating. I thought this manuscript would also be of interest to readers of Foods. However, a more detailed examination might be needed.
1. It is necessary to verify which part of the Sacha Inchi protein the identified peptides are cleaved from. If we search for Plukenetia volubilis in NCBI protein (https://www.ncbi.nlm.nih.gov/protein/), several amino acid sequences are registered. Are identified peptides cleaved from those proteins?
2. The authors have identified long-chain peptides such as SVMGPYYNSK, EWGGGGCGGGGVSSLR, RHWLPR, LQDWYDK, and ALEETNYELEK. They should be tested to see if they cannot be further cut with pepsin and pancreatin.
3. These peptides might be too long to absorb through the digestive tract. The authors should examine how these peptides act on the human body.
4. It is necessary to quantitatively analyze how much of these peptides are contained in 1g of Sach Inchi cake.
Author Response
Response to Reviewer 3 Comments
Thank you very much for taking the time to review our manuscript. Please find the detailed responses below and the corresponding revisions/corrections highlighted/in track changes in the re-submitted file.
- It is necessary to verify which part of the Sacha Inchi protein the identified peptides are cleaved from. If we search for Plukenetia volubilis in NCBI protein (https://www.ncbi.nlm.nih.gov/protein/), several amino acid sequences are registered. Are identified peptides cleaved from those proteins?
Answer: Thank you for bringing this point to our attention. One of the main limitations in identifying the source protein of these peptides is the scarce information available on the SI proteome. We recently published a paper that provides information on this topic (https://doi.org/10.3390/foods13203275). The subsequent analysis we propose is to adapt the reported methodology (find-pep-seq script) to profile these proteins.
- The authors have identified long-chain peptides such as SVMGPYYNSK, EWGGGGCGGGGVSSLR, RHWLPR, LQDWYDK, and ALEETNYELEK. They should be tested to see if they cannot be further cut with pepsin and pancreatin.
Answer: Thank you for bringing this point to our attention. These peptides are likely susceptible to cleavage by pepsin and pancreatin.
In this study, we aim to establish a methodology to identify (in silico) sequences after simulated gastrointestinal digestion (in vitro). We acknowledge that further studies are needed, in which these peptides will be synthesized and subjected to digestion to confirm cleavage.
- These peptides might be too long to absorb through the digestive tract. The authors should examine how these peptides act on the human body.
Answer: Yes. In fact, only two peptides (RHWLPR and ALEETNYELEK) out of the 20 analyzed have the potential to be cell-penetrating peptides (CPP), according to the results shown in Table 4. Again, we know about this limitation and hope to conduct further studies in cell models to evaluate the absorption of peptides.
- It is necessary to quantitatively analyze how much of these peptides are contained in 1g of Sach Inchi cake.
Answer: Thank you for bringing this point to our attention. Yes, clear information of the quantity of each potential bioactive peptide should be calculated. We thank this suggestion for our future studies.
Reviewer 4 Report
Comments and Suggestions for Authors
This article is novel, well written, and provides important findings. Minor issues need to be adress, indluding structure structural adjustments. The comments are provided below:
· The abstract and introduction sections lack more precise food applications of isolated and analyzed antioxidant peptides and the importance of their incorporation as functional food components or additives of variant health-promoting industries.
· It is mentioned that more than 1000 antioxidant peptides have been identified from food sources, like milk and animal and plant-based materials. Can you provide some examples of antioxidant peptides that are also isolated from different by-products or agro-waste?
· Line 93- stylistically unclear sentence - SIPC adequacy and Sacha Inchi Protein Concentrate (SPC) production were conducted using essentially the procedure reported. Also, could you specify working conditions for freeze-drying?
· Why were specifically ABTS and ORAC tests selected for the measurement of antioxidant activity?
· Line 147- could you specify what was the desired density for microphage culture?
· I found (Line 171) somewhat confusing, regarding the control being set at 100%. Could you elaborate on what the control is being expressed as, and how the results are being calculated?
· Line 177 – Consider rephrasing “protocol described” to improve grammatical flow.
· Line 266 T- Technical mistake, remove the latter.
· Results presented in Figure 3. seem a bit unclear. Maybe the resolution and the marks in the pictures could be improved or it could be added as a Table either in the text or as a Supplementary material?
· Result from sections 3.4. and 3.5. are presented in Tables 2-4, but the tables are not positioned directly below their corresponding paragraphs so they seem out of order. Consider making adjustments that could improve the structure of the article.
Author Response
Response to Reviewer 4 Comments
Thank you very much for taking the time to review our manuscript. Your comments have allowed us to improve the document. Please find the detailed responses below and the corresponding revisions/corrections highlighted/in track changes in the re-submitted file.
- The abstract and introduction sections lack more precise food applications of isolated and analyzed antioxidant peptides and the importance of their incorporation as functional food components or additives of variant health-promoting industries.
Answer: Thank you for bringing this point to our attention. We have included this important observation in the abstract (lines 16-17) and the introduction (lines 51-54).
- It is mentioned that more than 1000 antioxidant peptides have been identified from food sources, like milk and animal and plant-based materials. Can you provide some examples of antioxidant peptides that are also isolated from different by-products or agro-waste?
Answer: As recommended, this information has been included (lines 61-68).
- Line 93- stylistically unclear sentence - SIPC adequacy and Sacha Inchi Protein Concentrate (SPC) production were conducted using essentially the procedure reported. Also, could you specify working conditions for freeze-drying?
Answer: As recommended, this information has been included (lines 108-116).
- Why were specifically ABTS and ORAC tests selected for the measurement of antioxidant activity?
Answer: When the antioxidant activity of a sample is measured, at least two different methods based on different mechanisms of action should be used. We selected the ABTS and ORAC tests because they assess antioxidant activity through complementary mechanisms of action, as explained in the document (lines 289-290).
- Line 147- could you specify what was the desired density for microphage culture?
Answer: As suggested, the phrase has been clarified (lines 167-168).
- I found (Line 171) somewhat confusing, regarding the control being set at 100%. Could you elaborate on what the control is being expressed as, and how the results are being calculated?
Answer: The clarification has been made, and the formula for this calculation has been inserted (lines 179).
- Line 177 – Consider rephrasing “protocol described” to improve grammatical flow.
Answer: As suggested, the phrase has been clarified (line 198).
- Line 266 T- Technical mistake, remove the latter.
Answer: Thank you for pointing out the error; we have corrected it (line 292).
- Results presented in Figure 3. seem a bit unclear. Maybe the resolution and the marks in the pictures could be improved or it could be added as a Table either in the text or as a Supplementary material?
Answer: We believe it is an image that summarizes and integrates the data obtained. We have included this image in its highest resolution in the document for a clearer reading of the results.
- Result from sections 3.4. and 3.5. are presented in Tables 2-4, but the tables are not positioned directly below their corresponding paragraphs so they seem out of order. Consider making adjustments that could improve the structure of the article.
Answer: We have organized the paragraphs, tables and figures accordingly (lines 294-400, 225-235, and 257-470, respectively).
Round 2
Reviewer 1 Report
Comments and Suggestions for Authors
This manuscript titled “Antioxidant Peptides from Sacha Inchi Meal: An In vitro, Ex vivo, and In silico Approach” (Manuscript ID: foods-3302397) examined the antioxidant activity of the peptides released from Sacha Inchi protein during gastrointestinal digestion by integrating in vitro, ex vivo, and in silico methods. It is interesting work for the fine processing of Sacha Inchi protein which is an oil-producing by-product. Authors have revised their manuscript according to reviewer’s comments, and right now no more comments are addressed.
Reviewer 3 Report
Comments and Suggestions for Authors
You should explain where you made changes in response to the reviewer’s comments.
In addition, at least one of the points raised should be supplemented with experimental data.
I advised the editorial team to wait long enough to conduct additional experiments.
